# Issues related to the retrieval of stratospheric aerosol particle size information based on optical measurements

Christian von Savigny[1] and Christoph G. Hoffmann[1]

[1]Institute of Physics, University of Greifswald, Felix-Hausdorff-Str. 6, 17489 Greifswald, Germany

**Correspondence:** C. von Savigny (csavigny@physik.uni-greifswald.de)

**Abstract.** Stratospheric sulfate aerosols play an important role for the physics and chemistry of the atmosphere. **The radiative and chemical effects of stratospheric sulfate aerosols depend critically on the aerosol particle size distribution and its variability. Despite extensive research spanning several decades, the scientific understanding of the particle size distribution of stratospheric aerosols is still incomplete.** Particle size **estimates (often represented by the median radius of an assumed mono-modal log-normal distribution with fixed width or by the effective radius)** reported in different studies cover a wide range – even under background stratospheric conditions – and particle size **estimates** retrieved from satellite solar occultation measurements in the optical spectral range show a tendency to be systematically larger than retrievals based on other optical methods. In this contribution we suggest a potential reason for these systematic differences. Differences between the actual aerosol particle size distribution and the size distribution **function** assumed for aerosol size retrievals may lead to systematic differences in retrieved aerosol **size estimates**. We demonstrate that these systematic differences may differ significantly for different measurement techniques, which is related to the different sensitivities of these measurement techniques to specific parts of the aerosol particle population. In particular, stratospheric aerosol size retrievals based on solar occultation observations may yield systematically larger particle size estimates **(median or effective radii)** compared to, e.g., lidar backscatter measurements. Aerosol concentration – on the other hand – may be systematically smaller in retrievals based on occultation measurements compared to lidar measurements. **The results indicate that stratospheric aerosol size retrievals based on occultation or lidar measurements have to be interpreted with caution, as long as the actual aerosol particle size distribution is not well known.**

## 1 Introduction

According to Robock (2015) the variation of the stratospheric aerosol particle size as a consequence of volcanic injections into the stratosphere is one of the key open science questions of current stratospheric aerosol research. Considering the relatively large differences between stratospheric aerosol particle size **estimates** reported in the limited number of available studies (McLinden et al., 1999; Bingen et al., 2003; Deshler et al., 2003; Bingen et al., 2004a; Bingen at al., 2004b; Bourassa et al., 2008; Deshler, 2008; Malinina et al., 2018; Ugolnikov et al., 2018; Zalach et al., 2019), it appears fair to state that particle size parameters of stratospheric aerosols even under background conditions **are not well established**. For example, stratospheric aerosol particle size retrievals – for volcanically quiescent periods – from satellite occultation measurements tend to yield

relatively large median radii of up to several hundred nm (e.g., Bingen et al., 2003, 2004a; Bingen at al., 2004b; Wrana, 2019; Wrana et al., 2020), whereas retrievals based on several other measurement techniques typically lead to significantly smaller median radii (e.g., McLinden et al., 1999; Bourassa et al., 2008; Malinina et al., 2018; Ugolnikov et al., 2018; Zalach et al., 2019).

One of the main issues of optical techniques to investigate sizes of aerosol particles is the strong dependence of the aerosol scattering cross section on wavelength. In the Rayleigh limit (i.e., Mie size parameters of $\alpha \ll 1$) the scattering cross section scales with the 6th power of the particle radius. For larger Mie size parameters the dependence is weaker. Still, the signal observed by optical instruments (which may, e.g., take measurements of aerosol extinction in transmission or aerosol scattering in case of satellite limb-scatter or ground-based lidar measurements) will only be weakly affected by the smaller particles. This

effect is well known. Wurl et al. (2010) attempted to address this "blind spot issue" by adding a priori information on the population of small particles contributing very little to the measured optical signal.

    The differences in retrieved stratospheric aerosol particle size **estimates** between the different observation techniques and the relatively large size **estimates** routinely retrieved from solar occultation measurements may have different reasons. They could be related to measurement errors or to atmospheric variability. Both explanations, however, appear implausible. The first one,

because particle size retrievals from measurements with different solar occultation instruments (SAGE II, SAGE III/Meteor-3M and SAGE III/ISS) yield similar results (Bingen et al., 2003; Wrana, 2019; Wrana et al., 2020). The second explanation can be excluded by analyzing large data sets. The differences may also be caused by erroneous assumptions made in the retrievals in combination with different intrinsic sensitivities of the different measurement techniques to different fractions of the aerosol particle population. This potential reason is the topic of the current study. In this context the following aspects are important:

(a) stratospheric aerosol particle size retrievals from satellite transmission or scattering measurements are typically based on a mono-modal log-normal particle size distribution; (b) balloon-borne in-situ particle size measurements (e.g., Deshler et al., 2003; Deshler, 2008) usually yield bi-modal particle size distribution**s** with a main mode with a median radius on the order of 80 – 100 nm and a larger mode at median radii of typically about 400 nm. The larger mode is enhanced during periods with volcanically elevated stratospheric aerosol levels, but is also present during volcanically quiescent periods.

The main idea behind the present study is to simulate lidar backscatter and solar occultation measurements for a bi-modal aerosol particle size distribution and to retrieve particle size information assuming a mono-modal log-normal size distribution with fixed width parameter. This is done in order to investigate potential intrinsic differences in systematic retrieval errors between different measurements geometries.

    **Note that we do not claim that the actual particle size distribution of stratospheric aerosols is a bi-modal log-normal**
**distribution. This is an assumption, but an assumption based on several decades of in-situ particle size measurements (e.g., Deshler et al., 2003; Deshler, 2008).**

## 2 Methodology

Aerosol particle size information can in principle be obtained based on measurements of (a) the spectral dependence of the aerosol extinction or scattering coefficients (e.g., Yue and Deepak, 1983; Bingen et al., 2003), **(b) the spectral dependence of radiance, e.g., in limb geometry (e.g., Rieger et al., 2014; Malinina et al., 2018)**, **(c)** the scattering phase function (e.g., Gumbel et al., 2001; Renard et al., 2008), or **(d)** the polarization of the radiation scattered by aerosols (e.g., McLinden et al., 1999). In the present study the spectral method (a) is applied to forward simulations with a Mie-scattering code for different observation geometries frequently used to remotely sense stratospheric aerosols, i.e., satellite based occultation (stellar, lunar or solar) measurements and lidar measurements.

Although satellite occultation instruments and ground-based lidar instruments typically take measurements at different wavelengths – e.g., 1020 nm, 525 nm and 385 nm for SAGE II (Mauldin et al., 1985) compared to 1064 nm, 532 nm and 355 nm for the ALOMAR Rayleigh-Mie-Raman lidar (Langenbach et al., 2019; Zalach et al., 2019) – we assume for simplicity the same pair of wavelengths for all observation geometries studied here, i.e., $\lambda_1 = 1064$ nm and $\lambda_2 = 532$ nm.

The particle size distribution of stratospheric aerosols is **often** assumed to be well represented by a log-normal distribution:

$$\frac{dn}{dr}(r, r_m, S, N) = \frac{N}{\sqrt{2\pi}\ln(S)r} \cdot exp\left(-\frac{(\ln(r) - \ln(r_m))^2}{2\ln^2(S)}\right) \tag{1}$$

where $dn(r, r_m, S, N)$ corresponds to the number of particles in the $[r, r + dr]$ radius range per unit volume, $S$ represents the geometric standard deviation (distribution width), $r_m$ the median radius and $N$ the total particle concentration.

The basic idea behind the current study is to simulate the spectral dependence of stratospheric aerosol extinction and backscatter assuming a bi-modal log-normal distribution and based on Mie-scattering calculations for pure sulfate aerosol particles. These forward simulations are then used to retrieve aerosol particle size information assuming a mono-modal log-normal particle size distribution based on the spectral approach (a) mentioned above.

The existence of non-sulfate components of stratospheric aerosols is well established (e.g., Murphy et al., 1998; Curtius et al., 2005; Renard et al., 2008), but these particles are assumed to be of negligible impact in the current study.

For the occultation geometry the colour ratio $C_{occ}$ employed for the size retrievals is simply given by the ratio of the simulated Mie extinction cross sections $\sigma_{ext}(\lambda, r_m, S)$ at the two wavelengths:

$$C_{occ} = \frac{\sigma_{ext}(\lambda_1, r_m, S)}{\sigma_{ext}(\lambda_2, r_m, S)} \tag{2}$$

For the lidar geometry the colour ratio $C_{lid}$ is given by the ratio of the differential Mie scattering cross sections. The differential scattering cross section is given by product of the scattering cross section $\sigma_{sca}(\lambda, r_m, S)$ and the phase function $\Phi(\pi)$ for a scattering angle of $\Theta = \pi$. Note that the imaginary part of the extinction cross section for stratospheric sulfate aerosols can be neglected, implying $\sigma_{ext}(\lambda, r_m, S) = \sigma_{sca}(\lambda, r_m, S)$.

The colour ratio $C_{lid}$ is then given by:

$$C_{lid} = \frac{\sigma_{ext}(\lambda_1, r_m, S) \times P(\Theta = \pi, \lambda_1, r_m, S)}{\sigma_{ext}(\lambda_2, r_m, S) \times P(\Theta = \pi, \lambda_2, r_m, S)} \tag{3}$$

Stratospheric aerosol particle size information is also retrieved from limb-scatter measurements with the OSIRIS (e.g., Bourassa et al., 2008) and SCIAMACHY (e.g., Malinina et al., 2018) instruments. The limb-scatter geometry is not considered in the present study, because it requires accurate treatment of surface reflection and multiple scattering, which is beyond the scope of the treatment used here. The effects studied here should, however, also be investigated for aerosol size retrievals from limb-scatter measurements in future studies. **We would like to point out that different aspects of the sensitivity of satellite limb-scatter measurements on the stratospheric aerosol particle size distribution have been already investigated in the studies by Rieger et al. (2014) and Malinina et al. (2019).**

In the Rayleigh-limit – i.e., for very small particles – the scattering cross section scales with the 6th power of the particle radius. For larger particle sizes the scaling occurs with a smaller power. This is illustrated in Fig. 1. The top panel of the Fig. shows the dependence of the Mie extinction cross section (solid lines) on median radius of a mono-modal log-normal particle size distribution with a width of $S = 1.4$ and $N = 1$ at 532 nm and 1064 nm wavelength. Also shown (dashed lines) is the radius dependence of the differential scattering cross section for a lidar backscatter geometry, i.e., a scattering angle of $\Theta = \pi$. The bottom panel of Fig. 1 displays the exponent $\kappa$ of the power law, with which the extinction cross section scales, i.e., $\sigma_{ext} \propto r_m^\kappa$. The exponent $\kappa$ was determined using the following relation

$$\kappa(r_m) = \frac{\ln \frac{\sigma_{ext}(\lambda, r_m)}{\sigma_{ext}(\lambda, r_m + \Delta r_m)}}{\ln \frac{r_m}{r_m + \Delta r_m}} \tag{4}$$

According to Fig. 1 the extinction cross section scales with the 4th to 5th power of the median radius for typical median radii of about 100 nm.

**The Laramie in-situ measurements (e.g., Deshler et al., 2003; Deshler, 2008) typically indicate a bi-modal particle size distribution with a small mode with median radii on the order of 100 nm and a larger mode with median radii of several hundred nm. In the following we will model a bi-modal stratospheric aerosol size distribution by superposing two mono-modal log-normal distributions with different median radii and different width parameters:**

$$\frac{dn_{tot}}{dr}(r, r_{m,1}, r_{m,2}, S_1, S_2, N_1, N_2, \chi) = \frac{\frac{dn}{dr}(r, r_{m,1}, S_1, N_1) + \chi \cdot \frac{dn}{dr}(r, r_{m,2}, S_2, N_2)}{\chi + 1} \tag{5}$$

where $\chi$ is the so-called *coarse mode fraction* (CMF).

Figure 2 shows the bi-modal particle size distribution (black solid line) with two log-normal modes and the following size parameters: $r_1 = 80$ nm, $S_1 = 1.6$, $N_1 = 1$, $r_2 = 400$ nm, $S_2 = 1.15$, $N_2 = 1$ and a coarse mode fraction of $\chi = 10^{-2}$. These values are approximately consistent with the available in-situ measurements (Deshler et al., 2003; Deshler, 2008). The red dashed line displays the particle size distribution multiplied by the Mie-extinction cross section $\sigma_{ext}(\lambda, r)$ and describes the contribution

of the different aerosol sizes to the overall extinction seen by an occultation instrument. Apparently, the importance of the second mode is enhanced and also for the main mode it is obvious that the larger particles contribute more to the overall extinction compared to the smaller particles. The blue dash-dotted line corresponds to the particle size distribution multiplied by the differential scattering cross section for a lidar backscatter geometry, i.e., a scattering angle of $\Theta = \pi$. Apparently, aerosol particles with sizes close to the main mode contribute most to the backscatter signal measured by a lidar. Figure 2 illustrates qualitatively, that occultation measurements will be more sensitive to larger particles of the aerosol population than the lidar backscatter measurements. **This already suggests that size retrievals based on occultation measurements may lead to larger estimates of the aerosol size, if a wrong size distribution function is assumed. We note that the width of the small mode of the bi-modal size distribution has been chosen larger than for the simulations described below – in order to illustrate the effect more clearly.**

The basic procedure used here is to simulate the colour ratios for the different observation geometries (see Eqs. (2) and (3)) assuming the bi-modal log-normal particle size distribution, followed by the retrieval of the median radius of an assumed mono-modal log-normal size distribution with fixed geometric width S. **Note that we also discuss results for the retrieval of the effective radius below (see section 3.1).** For the size retrievals precalculated look-up-tables (LUT) of the colour ratios as a function of median radius are employed. The retrieved median radii are determined from the forward simulated colour ratios by linear interpolation using the LUTs. The following parameters are assumed for the forward simulations: $r_1$ = 80 nm, $S_1$ = 1.4, $N_1$ = 1, $r_2$ = 400 nm, $S_2$ = 1.15, $N_2$ = 1, guided by the available in-situ measurements (Deshler et al., 2003; Deshler, 2008). **Note that the exact values of the assumed bi-modal size distribution parameters are not important for the main conclusion of the present study**. It is also important to mention that the size retrieval based on multi-color lidar measurements does not necessarily yield a unique solution. This aspect was discussed in detail by Zalach et al. (2019) and does not affect the main conclusions of the current investigation. For the size retrievals performed here, the geometric width of the log-normal distribution was chosen to be identical to the geometric width of the main mode of the bi-modal particle size distribution used for the forward simulations. For the analysis of observational data sets the correct value of the distribution width is certainly not available, but this assumption does not affect the main conclusions of the present study, either.

## 3 Results and discussion

### 3.1 Particle size retrievals

**We first discuss results for the retrieval of the median radius of the assumed mono-modal log-normal particle size distribution, followed by a discussion of the effects on the retrieval of the effective radius.** Figure 3 displays the dependence of the retrieved median radius for the lidar (red line) and occultation geometries (blue line) as a function of coarse mode fraction $\chi$. For very small coarse mode fractions (i.e., $\chi < 10^{-4}$) the retrieved median radii are in good agreement with the median radius of the main mode, if the correct width of the particle size distribution is assumed. For increasing values of the coarse mode fraction the median radius based on the occultation measurements increases significantly and reaches about 3.5 times the median radius of the main mode for a coarse mode fraction of $\chi = 10^{-1}$. This strong overestimation of the retrieved

size for occultation measurements relative to the lidar retrievals is directly related to the higher sensitivity of the occultation measurements to the larger particles (see Fig. 2) – compared to the lidar measurements – in combination with neglecting the second mode for the retrievals.

The coarse mode fractions reported in different studies are on the order of $10^{-3} - 10^{-2}$, corresponding to median radii retrieved from occultation measurements of up to about 300 nm according to Fig. 3. In other words, the simulated size retrievals described above not only qualitatively reproduce the larger stratospheric aerosol particle sizes frequently reported in studies based on occultation measurements, the sizes are also reproduced quantitatively, based on bi-modal particle size distribution parameters reported in previous studies.

Next we test how retrievals of the effective radius for the lidar and occultation measurement geometries depend on the coarse mode fraction of the bi-modal particle size distribution. The results for the effective radius are shown in Fig. 4 in a similar way as for the median radius in Fig. 3. The effective radii for the lidar (red solid line) and occultation (blue solid line) retrievals were determined from the median radii and the assumed distribution width using the following relationship (Grainger, 2017):

$$r_{eff} = r_m \exp\left(\frac{5}{2}\ln^2 S\right) \tag{6}$$

The effective radius of the bi-modal log-normal distribution (black dashed line in Fig. 4) was determined by numerical integration. For the parameters considered here, the maximum overestimation of the effective radius of the occultation retrievals – relative to the true effective radius – occurs for a coarse mode fraction of about $6 \times 10^{-3}$. For this coarse mode fraction the effective radius retrieved from the simulated occultation measurements is about a factor of 2 larger than the true value and a factor of 3 larger than the value retrieved from the simulated lidar backscatter measurements.

Another important parameter in aerosol research is the Ångström exponent or spectral exponent $\alpha$, which corresponds to the exponent of a power law used to approximate the spectral dependence of the aerosol extinction cross section: $\sigma_{ext}(\lambda) \propto \lambda^{-\alpha}$. Based on the simulated aerosol extinction cross sections, the Ångström exponent is easily determined using the following relationship: $\alpha = \ln(\sigma_{ext}(\lambda_1)/\sigma_{ext}(\lambda_2))/\ln(\lambda_2/\lambda_1)$ with $\lambda_1$ = 1064 nm and $\lambda_2$ = 532 nm in this study. The dependence of $\alpha$ on the coarse mode fraction of the assumed bi-modal log-normal distribution is shown in Fig 5. As expected, $\alpha$ is smaller for larger coarse mode fractions and asymptotically approaches a larger value for decreasing coarse mode fractions. The exact dependence of $\alpha$ on the coarse mode fraction certainly depends on the specific parameters of the assumed bi-modal particle size distribution, but the overall effect is similar.

## 3.2 Aerosol number density retrievals

In a similar way as for the retrieved aerosol median radius we tested the dependence of the retrieved aerosol number density on the coarse mode fraction of the assumed bi-modal log-normal particle size distribution if only a single mode is considered in the retrievals. It is to be expected that the lidar retrievals will overestimate the aerosol number density and the occultation retrievals will tend to underestimate it. Figure 6 displays the dependence of the retrieved aerosol number densities as a function of the coarse mode fraction $\chi$. For the smallest coarse mode fractions the true aerosol number density can be retrieved accurately, as

expected. For increasing coarse mode fraction the aerosol density retrieved for the occultation geometry deviates significantly from the true value and for $\chi = 10^{-2}$ the retrieved number density is only about 5% of the true value. At a typical coarse mode fraction value of $\chi = 10^{-2}$ the ratio of the aerosol densities retrieved from the lidar and occultation measurements is roughly 50.

The results clearly show that substantial errors in retrieved aerosol number density have to be expected, if the assumed aerosol particle size distribution differs from the actual distribution. This is expected and not surprising. The important point is that number density retrievals based on solar occultation measurements may systematically underestimate the aerosol number density, while lidar measurements may lead to a systematic overestimation. For the range of coarse mode fractions reported in the literature (e.g., Deshler, 2008; Chen et al., 2018), the number densities retrieved from occultation and lidar measurements may differ by more than one order of magnitude. Results published in earlier studies are qualitatively and semi-quantitatively consistent with these conclusions (compare, e.g., the aerosol number densities retrieved from occultation measurements in Bingen et al. (2003) with the ones retrieved from lidar measurements by Zalach et al. (2019)).

### 3.3 Surface area and volume density

For mono-modal log-normal distributions surface area density (SAD) and volume density (VD) can be determined via the following analytical formulae (see, e.g., Grainger, 2017):

$$SAD = 4\pi N r_m^2 \exp\left(2\ln^2 S\right) \tag{7}$$

$$VD = \frac{4}{3}\pi N r_m^3 \exp\left(\frac{9}{2}\ln^2 S\right) \tag{8}$$

The variables in Eqs. (7) and (8) are the same as in Eq. (1). The surface area and volume densities are determined using the retrieved median radius and number density for each of the considered observation geometries. The resulting surface area and volume densities are compared in the left panels of Fig. 7 and Fig. 8, respectively. The black dashed lines in the left panels of the Figs. correspond to the true values – as a function of coarse mode fraction – based on the assumed bi-modal log-normal particle size distribution. The right panels of Fig. 7 and 8 show the ratios of the retrieved values to the true value. For both quantities the correct values can essentially be retrieved for the smallest coarse mode fractions if the correct width of the particle size distribution is assumed. For coarse mode fractions between $10^{-2}$ and $10^{-1}$ the relative differences between the retrieved SADs and the true values can reach a factor of about 4 and the ratio between the SAD retrieved from lidar measurements and the one retrieved from occultation measurements reaches one order of magnitude. The differences are slightly smaller for the volume density (see right panel of Fig. 8). Considering Eqs. (7) and (8), this behavior is consistent with the coarse mode fraction dependence of median radius (see Fig. 3) and aerosol number density (see Fig. 6). Note that the relative retrieval errors are smaller for surface area density and volume density compared to median radius or aerosol number density. This is because the retrieval errors in median radius and number density partially compensate each other when calculating surface area density and volume density.

| Case | $r_1$ | $S_1$ | $r_2$ | $S_2$ |
|------|-------|-------|-------|-------|
| 1 | 80 nm | 1.4 | 400 nm | 1.15 |
| 2 | 50 nm | 1.4 | 300 nm | 1.2 |
| 3 | 110 nm | 1.4 | 350 nm | 1.2 |

Table 1. Overview of parameters of the bi-modal particle size distributions assumed for the forward simulations.

### 3.4 Extinction retrievals from lidar measurements

Finally we discuss the effects on aerosol extinction coefficients retrieved from the simulated lidar backscatter measurements. Since the imaginary part of the refractive index of the stratospheric aerosols is assumed to be zero, scattering and extinction coefficients are identical. The extinction coefficient retrieved from the simulated lidar measurements is easily determined based on the retrieved aerosol median radius – which is employed to calculate the extinction cross section using the Mie routine – and the number density determined above. Figure 9 shows the dependence of the ratio of the retrieved and true extinction coefficient on the coarse mode fraction for the two wavelengths considered, i.e., 532 nm and 1064 nm. For small coarse mode fractions the true extinction coefficient can be retrieved very well, as expected. However, as the coarse mode fraction increases, the retrieved extinction coefficient systematically underestimates the true value for both wavelengths. This behaviour is to be expected, because of the lower sensitivity of the lidar measurements to the larger fraction of particles compared to the smaller particles of the assumed bi-modal particle size distribution (see Fig. 2). As Fig. 9 shows, the low bias of the retrieved extinction coefficient compared to the true value is larger for the longer wavelength. At at wavelength of 1064 nm the retrieved extinction is only about 20 % of the true value for a coarse mode fraction of $\chi = 10^{-1}$ – the largest value considered here. For 532 nm the retrieved extinction is about 80 % of the true value for a coarse mode fraction of $\chi = 10^{-1}$. The larger bias of the extinction at the longer wavelength is consistent with the underestimation of the effective radius (Fig. 4), which is associated with a stronger wavelength dependence of the extinction coefficient.

### 3.5 Discussion

The results presented above are based on specific parameters of the assumed bi-modal particle size distribution. The question arises as to how the results depend on the specific assumptions made. In order to test this we performed forward simulations and retrievals based on different parameters of the bi-modal particle size distribution. For illustration, we show particle size retrieval results for two different sets of the parameters $r_1$, $S_1$, $r_2$ and $S_2$ of the bi-modal particle size distributions. Table 1 lists the parameters of these additional two cases (cases 2 and 3) together with the parameters used above (case 1).

Figure 10 shows the coarse mode fraction dependence of the retrieved median radius for case 2 and Fig. 11 for case 3. The retrieval results certainly depend to some extend on the specific assumptions made, but the overall conclusions are not affected. We note again that the main aspect of this study does not lie in the specific values of aerosol particle size

**information retrieved for specific assumptions. The main aspect is that size retrievals based on different measurement techniques may lead to different results due to the different intrinsic sensitivities of these techniques.**

Most likely the actual particle size distribution of stratospheric aerosols is highly variable in space and time (see, e.g., the highly structured patterns in lidar volume backscatter coefficients in Langenbach et al. (2019)). In addition, not only bi-modal distributions, but a variety of distributions deviating from a mono-modal log-normal distribution have to be expected. Based on the results presented here, one cannot expect that particle size retrievals based on simplified assumptions will yield the true values. In addition, size retrievals based on different measurement techniques will be affected by different systematic errors, because of the different sensitivities to certain parts of the particle population.

In addition to the interpretation of solar occultation and multi-color lidar measurements, the results presented here are also of relevance for stratospheric aerosol extinction profile retrievals from single-wavelength lidar measurements and from satellite limb-scatter measurements. In both cases assumptions on the aerosol particle size distribution have to made. For single colour lidar measurements knowledge on the particle size distribution is required in order to determine the lidar-ratio and subsequently convert lidar backscatter measurements to aerosol extinction coefficients (e.g., Khaykin et al., 2017). For limb-scatter measurements the particle size distribution is required to translate the measured limb-scatter signal to aerosol extinction (e.g., von Savigny et al., 2015; Chen et al., 2018).

Stratospheric aerosol particle size retrievals from satellite limb-scatter measurements **may be affected** by similar issues as the occultation and lidar retrievals described here. Limb-scatter measurements were not discussed here in detail, because these particle size retrievals require correct treatment of the much more complicated radiative transfer (considering surface reflection and multiple scattering). Simplified tests using Eq. (3) as a forward model for scattering angles different from $\pi$, showed that results for the quantities analyzed here (median radius, **effective radius**, number density, surface area density and volume density) roughly fall in the range between the results for occultation and lidar measurements (results not shown). The underlying assumption behind these simplified retrievals is that only singly-scattered photons are considered. As mentioned above, additional investigations are required to quantify the effects discussed here for the limb-scatter geometry.

**It is to be expected that the systematic differences in retrieved aerosol size parameters for lidar and occultation retrievals will increase after major volcanic eruptions (e.g., Pinatubo), because then the second particle mode at radii of several hundred nm will be enhanced (e.g., Deshler, 2008). For smaller eruptions there is evidence for a temporal decrease in aerosol effective radius (Larry Thomason, pers. comm. and Wrana et al. (2020)), which may lead to smaller differences between aerosol size parameters retrieved from lidar and occultation measurements.**

Despite the described limitations of aerosol particle size retrievals from different types of measurements, we strongly believe that particle size retrievals are still useful, as long as all underlying assumptions are clearly stated. Limitations of the retrievals should be transparently and explicitly discussed. Accurate in-situ measurements of the particle size distribution of stratospheric aerosols with high size resolving capability in the sub 100 nm size range are urgently needed for (a) a better general under-standing of the nature of the particle size distribution and (b) to improve the capabilities to remotely sense stratospheric aerosol size information and aerosol extinction using optical measurements.

We would like to point out **again** that we do not claim that the actual particle size distribution of stratospheric aerosols is a bi-modal log-normal distribution. This study simply tests the effects of assuming a mono-modal log-normal particle size distribution on the retrievals, if the actual distribution is a bi-modal log-normal distribution.

**It is also important to note that the correct particle size parameters can in principle be retrieved from measurements in lidar backscatter or occultation observation geometry using the colour ratio approach employed here (neglecting issues related to potential non-uniqueness of the solutions), if the assumption of a mono-modal log-normal particle size distribution is correct and one of the size parameters, e.g., the width $\sigma$ is known.** This is also illustrated in Figs. 3 and 6 for the median radii and aerosol numbers densities and in Figs. 7 and 8 for surface area and volume densities, respectively. The Figs. show that if the coarse mode fraction becomes very small, the retrieved values will approach the true values. **In addition, it should be pointed out that the non-uniqueness of the solution is a potential problem with aerosol size information retrievals based on colour ratios, which has to be kept in mind when interpreting retrievals.**

**The results presented here are also of importance for model simulations of stratospheric aerosols – some of which model aerosol growth processes more or less explicitly (e.g., Kokkola et al., 2009) – because stratospheric aerosol particle size information retrieved from solar occultation measurements is used in several studies for comparison with the model results (e.g., Hommel, 2008).**

## 4    Conclusions

A fundamental intrinsic difficulty for retrieving particle size information of stratospheric sulfate aerosols from remote sensing measurements in the optical spectral range was investigated. Size retrievals are usually based on a mono-modal log-normal particle size distribution, while the actual size distribution may be different, e.g., a bi-modal log-normal distribution. In this study we investigated, how aerosol size retrievals – assuming a mono-modal particle size distribution – are affected if the actual distribution is bi-modal. Simulations were carried out for satellite occultation measurements and lidar backscatter measurements. Due to the different sensitivities of the different observation techniques to different parts of the particle population, the size retrievals from simulated occultation and lidar measurements behave quite differently. The occultation retrievals yield substantially larger median radii than the lidar retrievals – by up to a factor or 3 for the assumed scenarios – while the retrieved aerosol number densities are systematically lower for occultation retrievals – by up to almost two orders of magnitude – compared to the lidar retrievals. These findings may be a reason for the relatively large stratospheric aerosol **particle size estimates** usually retrieved from solar occultation measurements. Based on realistic assumptions on the bi-modal log-normal size distribution we are able to reproduce differences between published lidar and occultation retrievals both in a qualitative and quantitative sense. The presented results challenge our current knowledge of the size distribution of stratospheric aerosols, which is mainly based on optical measurements. Stratospheric aerosol size retrievals from occultation measurements are, however, still valuable in our opinion, because they do contain information on aerosol size. Future studies should attempt exploiting simultaneous and co-located measurements of the same air volume with different measurement techniques in order to provide more pieces of information on the particle size distribution of stratospheric aerosols.

*Acknowledgements.* This work was funded by the Deutsche Forschungsgemeinschaft (DFG, project LESAP (grant SA-1351/7) and project VolARC (grant SA-1351/10) of the DFG research unit VolImpact (FOR 2820)). We also acknowledge support by the University of Greifswald and thank the Earth Observation Data Group at the University of Oxford for providing the IDL Mie routines used in this study.

*Author contributions.* CvS outlined the project, implemented the method, carried out the simulations and retrievals with assistance by CH.
All authors discussed, edited and proofread the manuscript.

*Competing interests.* The authors declare that they have no conflict of interest.

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

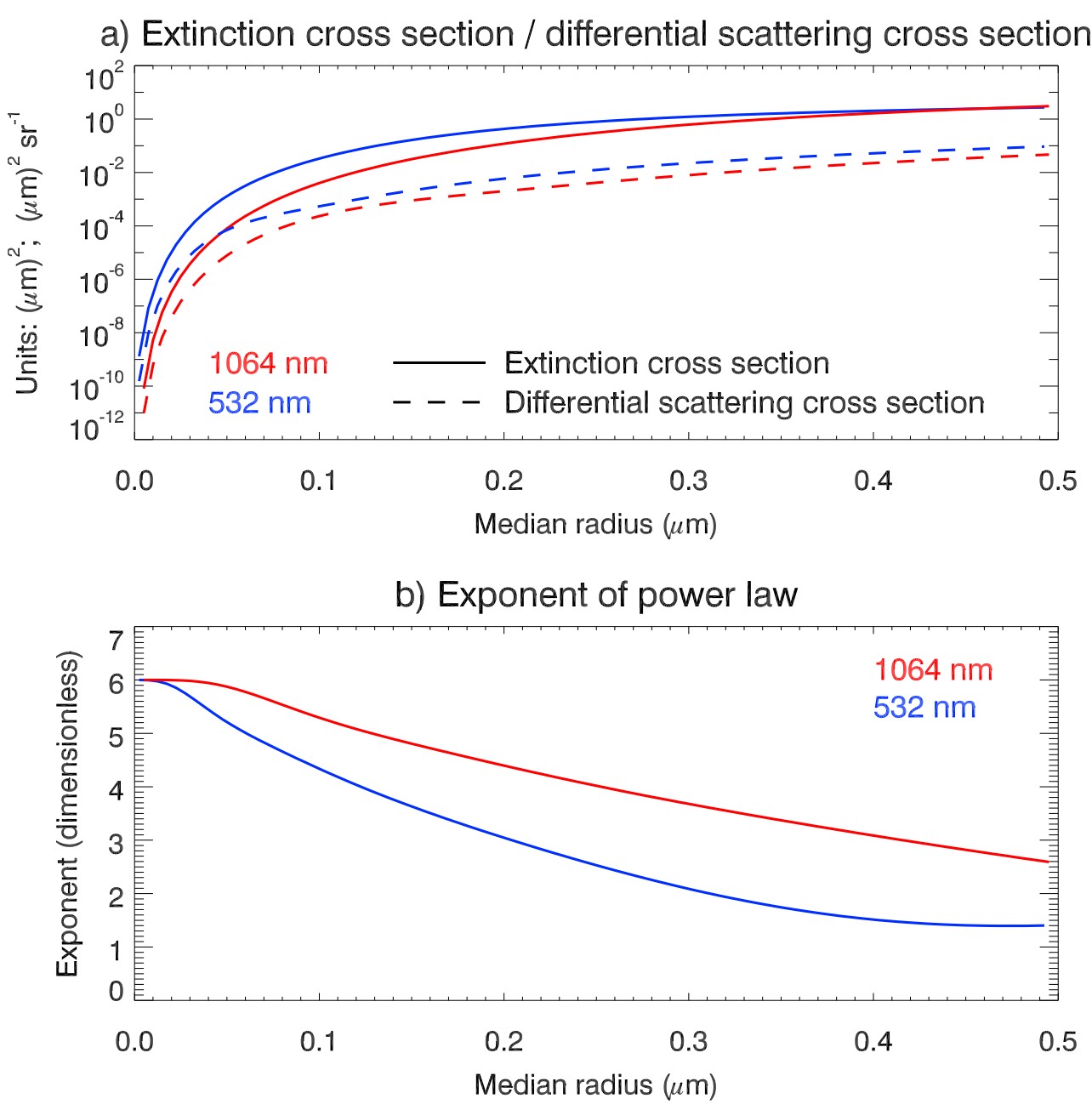

**Figure 1.** Top panel: The solid lines correspond to the median radius dependence of the Mie extinction cross section at 1064 nm (red line) and 532 nm (blue line) for a mono-modal log-normal particle size distribution with S = 1.4. The dashed lines display the median radius dependence of the differential scattering cross section for a lidar backscatter geometry. Bottom panel: Dependence of the power law exponent (see Eq. (4)) of the dependence of the aerosol extinction cross section on median radius.

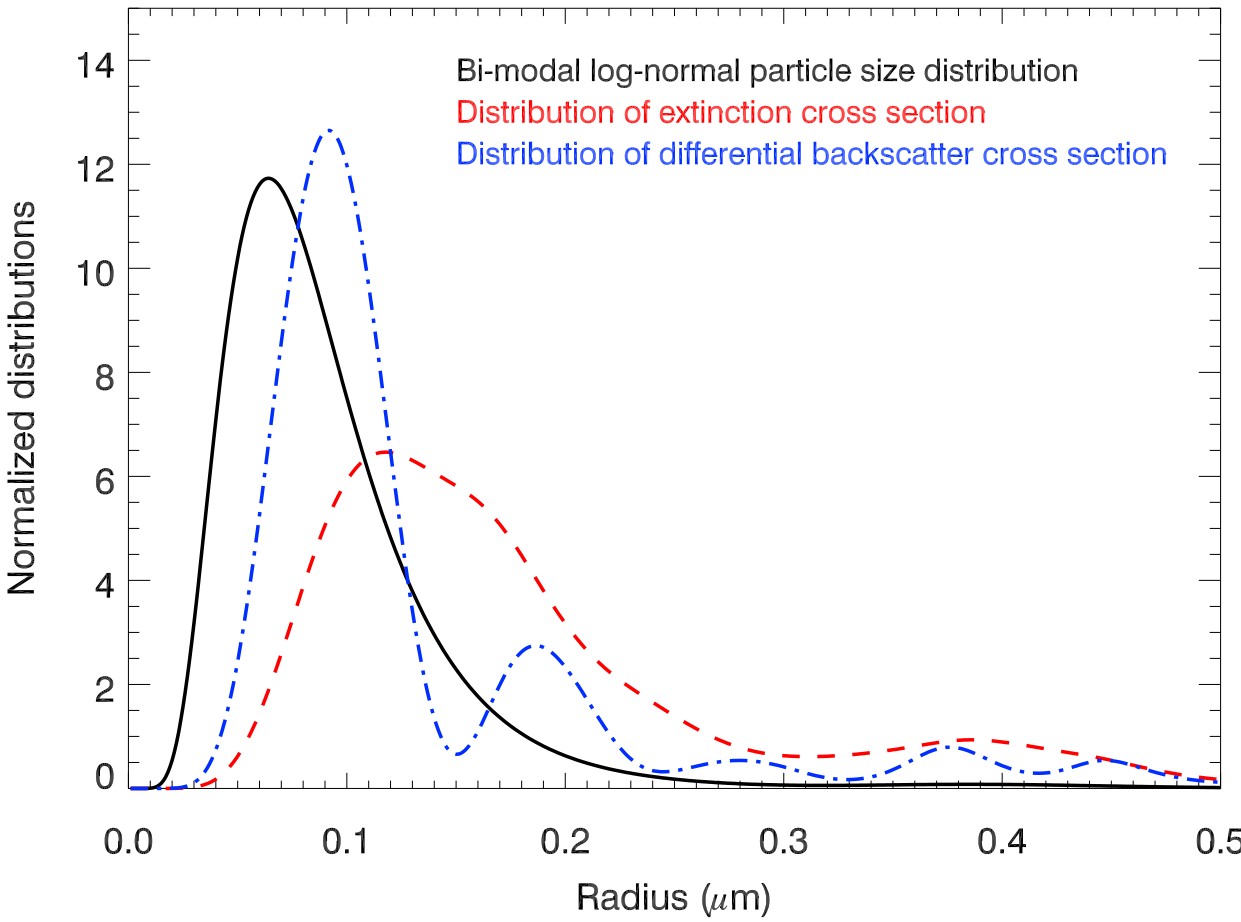

**Figure 2.** The black line shows the assumed bi-modal particle size distribution with two log-normal modes and the following parameters: $r_1 = 80\,\text{nm}$, $S_1 = 1.6$, $r_2 = 400\,\text{nm}$, $S_2 = 1.15$ and a coarse mode fraction of $\chi = 10^{-2}$. The red dashed line corresponds to the particle size distribution multiplied by the Mie-extinction cross section $\sigma_{ext}(\lambda = 532\,\text{nm}, r)$. The blue dash-dotted line corresponds to the particle size distribution multiplied by the differential scattering cross section for a lidar backscatter geometry and also for a wavelength of $532\,\text{nm}$. Note that all functions are normalized.

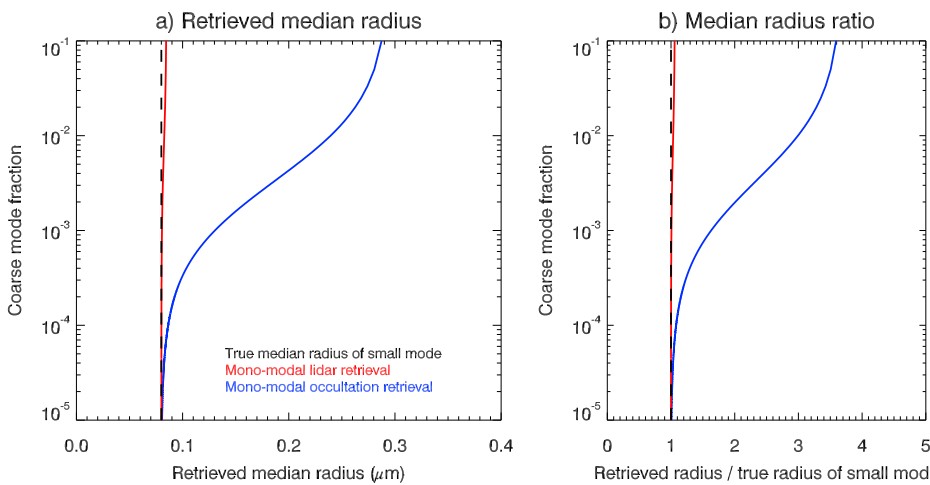

**Figure 3.** Left panel: The coarse mode fraction dependence of retrieved stratospheric aerosol median radius $r_m$ is shown for an assumed mono-modal log-normal particle size distribution with S = 1.4 based on simulated lidar (red line) and satellite occultation (blue line) measurements. The actual aerosol particle size distribution is a bi-modal log-normal distribution. The black dashed line corresponds to the true median radius of the small mode. Note that the distribution width of the mono-modal distribution assumed for the retrievals was chosen to be identical to the width of the main mode of the bi-modal log-normal distribution used for the forward simulations. For the smallest coarse mode fractions, the median radius of the small mode can be well retrieved for all measurement geometries, as expected. Right panel: Ratio of the retrieved radii and the true median radius of the small mode of the bi-modal log-normal distribution.

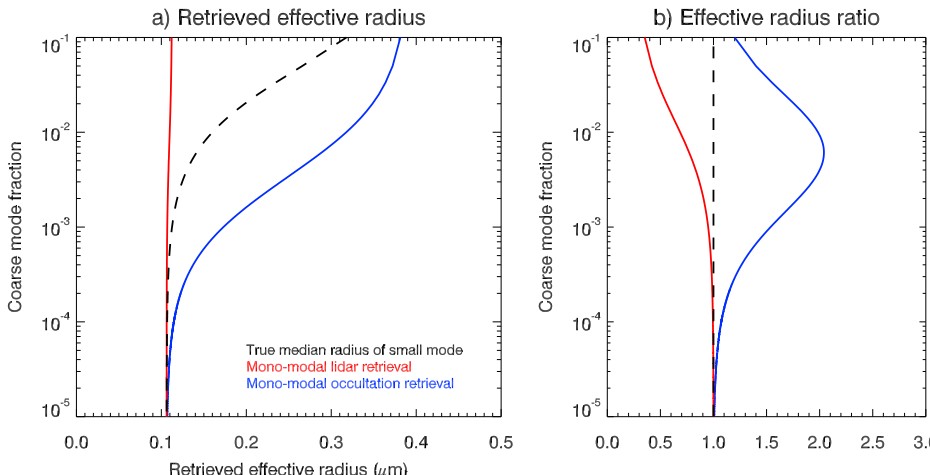

**Figure 4. Left panel: The coarse mode fraction dependence of retrieved effective radius $r_{eff}$ is shown for an assumed mono-modal log-normal particle size distribution with S = 1.4 based on simulated lidar (red line) and satellite occultation (blue line) measurements. The black dashed line corresponds to the true effective radius of the bi-modal particle size distribution assumed for the forward simulations. Right panel: Ratio of the retrieved effective radii and the true median radius of the bi-modal log-normal distribution.**

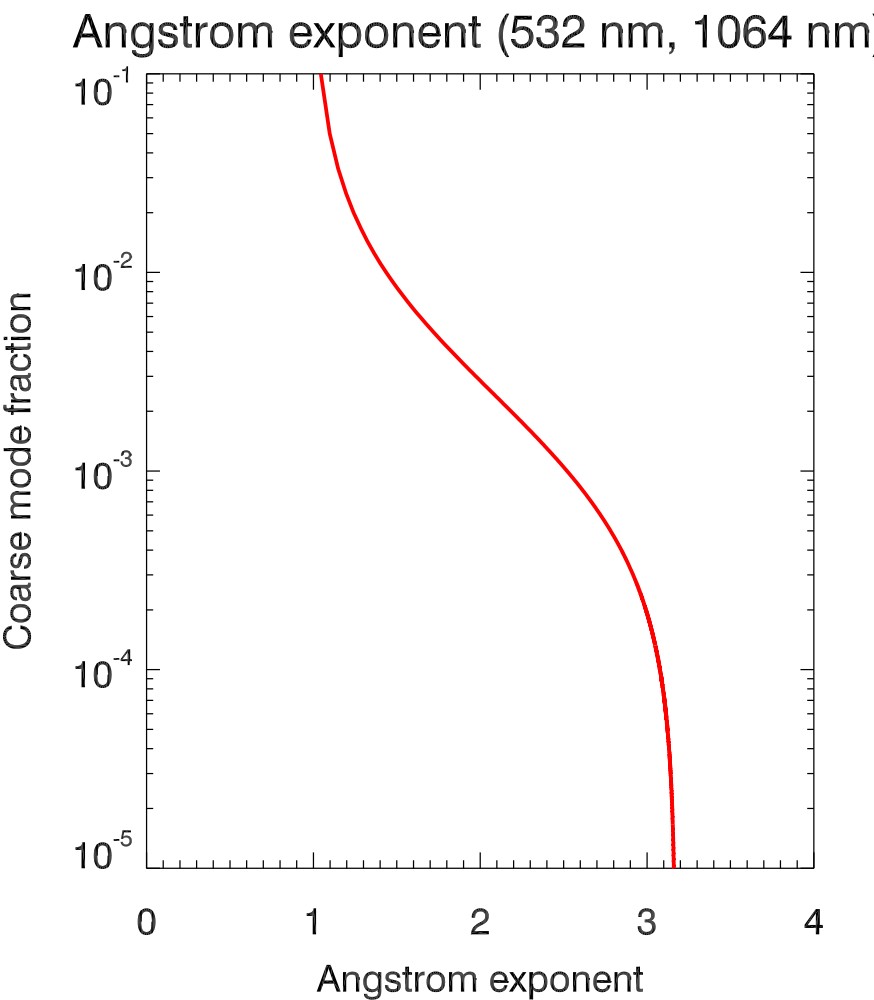

**Figure 5. Shown is the dependence of the Ångström exponent $\alpha$ – determined from the simulated extinction cross sections – on the coarse mode fraction of the assumed bi-modal particle size distribution.**

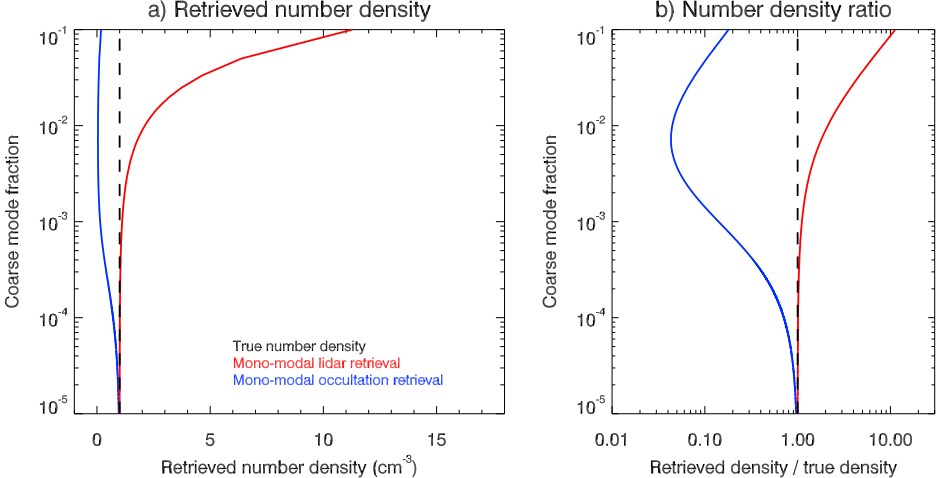

**Figure 6.** Similar to Fig. 3 but for aerosol number density.

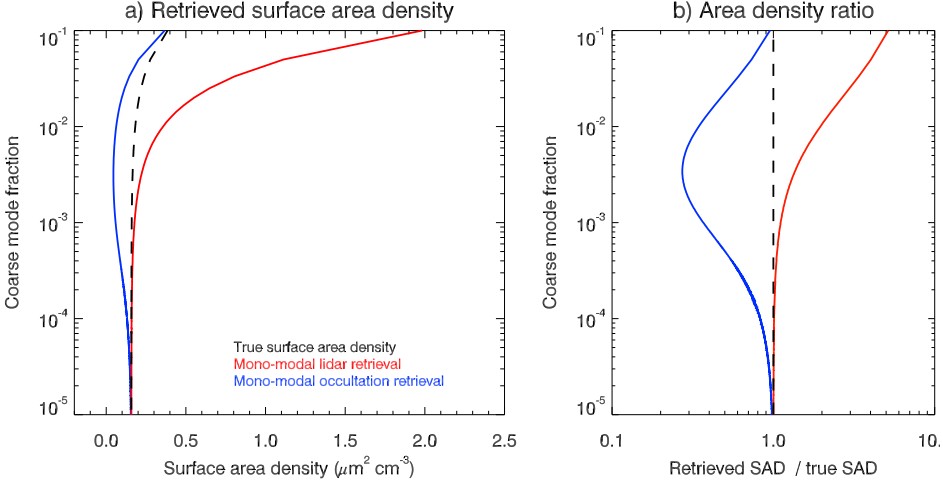

**Figure 7.** Similar to Fig. 3 but for surface area density.

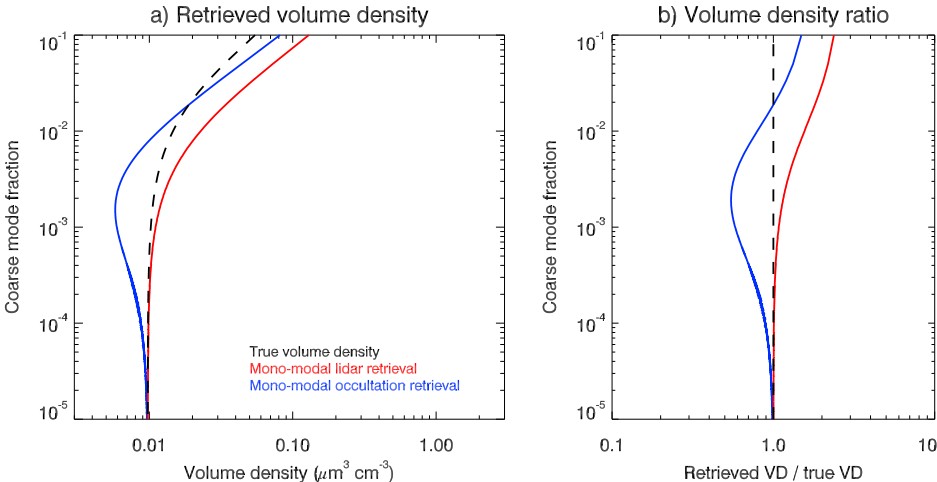

**Figure 8.** Similar to Fig. 3 but for volume density.

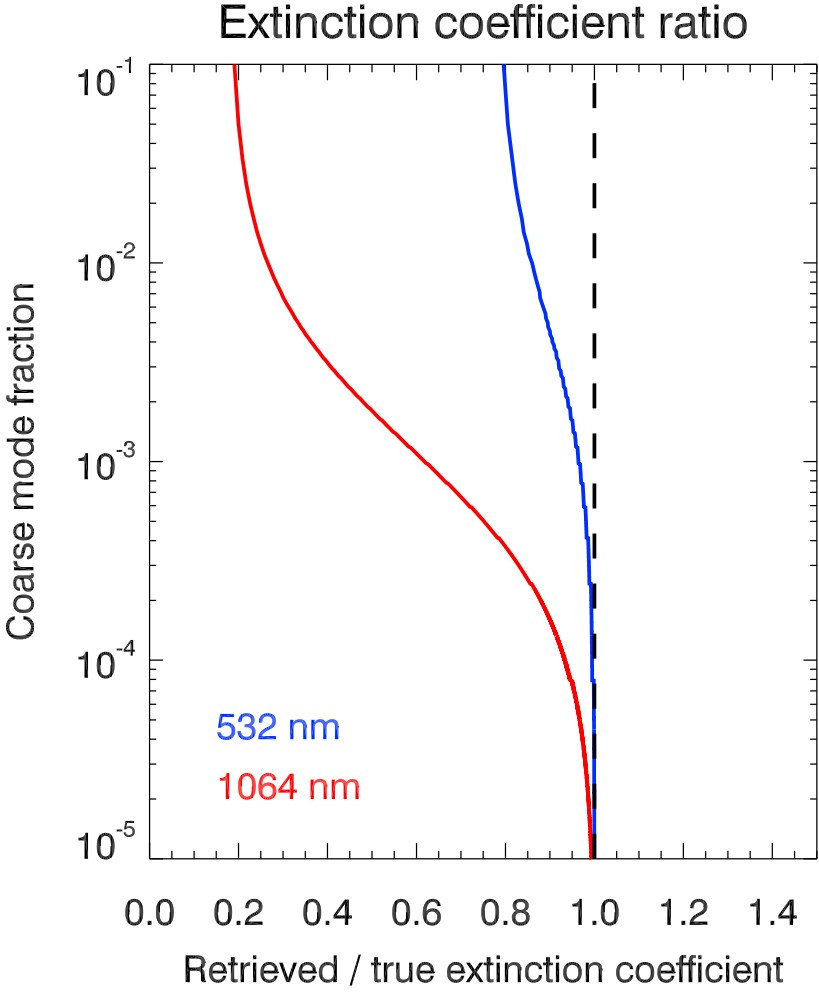

**Figure 9.** Shown is the ratio of the extinction coefficient retrieved from the simulated lidar measurements and the true extinction coefficient as a function of the coarse mode fraction of the bi-modal particle size distribution and for the two wavelengths considered. For decreasing coarse mode fractions the ratio asymptotically approaches 1 as expected.

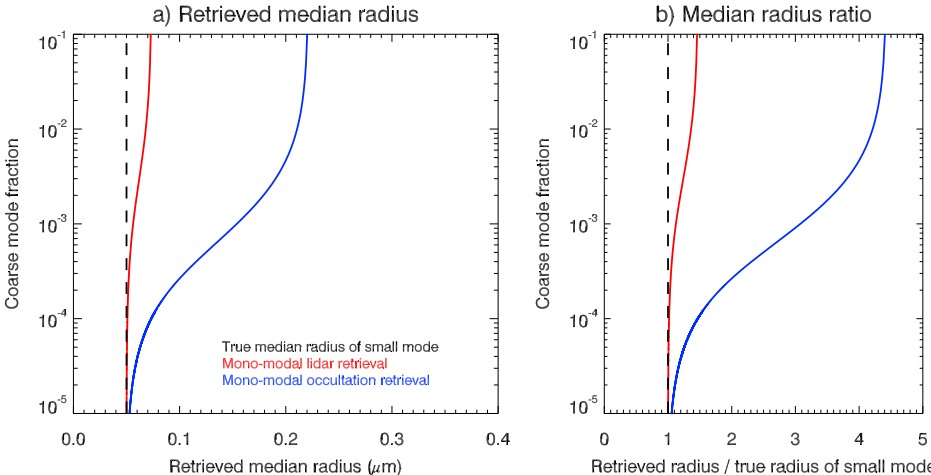

**Figure 10. Similar to Fig. 3, but for a different set of parameters of the bi-modal particle size distribution ($r_1$ = 50 nm, $S_1$ = 1.4, $r_2$ = 300 nm, $S_2$ = 1.2, i.e., case 2 in Table 1.**

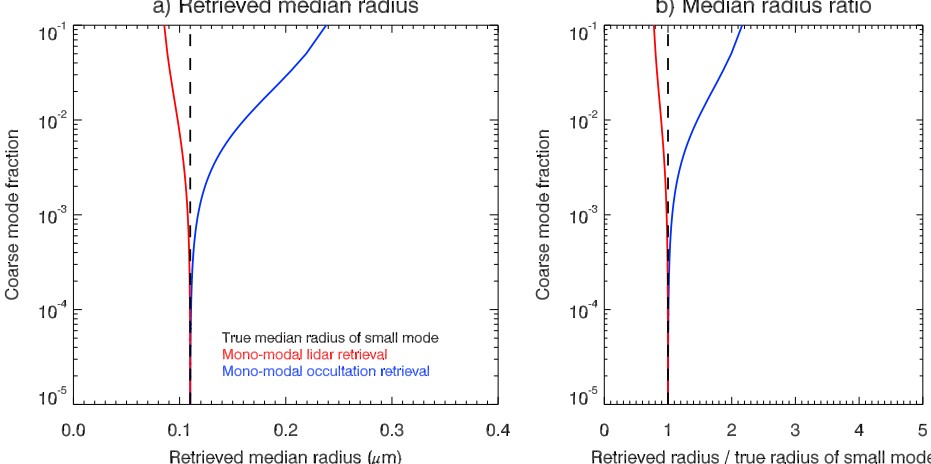

**Figure 11. Similar to Fig. 3, but for a different set of parameters of the bi-modal particle size distribution ($r_1$ = 110 nm, $S_1$ = 1.4, $r_2$ = 350 nm, $S_2$ = 1.2, i.e., case 3 in Table 1.**