# Peer review of "Issues related to the retrieval of stratospheric aerosol particle size information based on optical measurements"

_Atmospheric Measurement Techniques, 2019_

## Short Comment (SC1) · 26 Sep 2019

Dear Christian and Christoph,

In your paper you show very interesting results illustrating that a weaker sensitivity of the solar occultation measurements to smaller particles is expected to lead to the overestimation of the mode radius and underestimation of the particle number density if a coarse mode is present but not considered in the retrieval. However, you tend to over-generalize your conclusions. As shown by Malinina et al. (2019), not mentioned in your paper by the way, the sensitivity of the limb measurements to smaller particles is very different to that of solar occultation. For this reason I disagree with a blind extension

of your findings to the limb geometry as you do, e.g. by writing "Stratospheric aerosol particle size retrievals from satellite limb-scatter measurements can be expected to be affected by similar issues as the occultation and lidar retrievals described here. " To my opinion this statement is unjustified and should be removed.

Furthermore, with the last sentence of the abstract "The results question the overall significance of stratospheric aerosol size retrievals based on optical satellite or lidar measurements, as long as the actual aerosol particle size distribution is not well known." you provide a misleading message to the scientific community. First, based on the results of your paper you can only talk about solar occultation measurements and must not generalize your conclusions to all optical satellite methods, second a known bias in the retrieval products is not yet a reason to question the significance of the retrieval/measurements in general.

One more technical issue is in the first paragraph of the "Methotology" section "Aerosol particle size information can in principle be obtained based on measurements of (a) the spectral dependence of the aerosol extinction or scattering coefficients (e.g., Yue and Deepak, 1983; Bingen et al., 2003), (b) the scattering phase function (e.g., Gumbel et al., 2001; Renard et al., 2008), or (c) the polarization of the radiation scattered by aerosols (e.g., McLinden et al., 1999)." Here you seem to forget that in limb retrievals the spectral dependence of the radiance rather than that of the extinction or scattering coefficient is used (Malinina et al., 2018).

Kind regards,

Alexei

---

## Referee Comment (RC1) · Anonymous Referee #1 · 29 Oct 2019

The paper contains some useful information. But overall content of the paper is weak and would be of limited interest to the researchers in the field. Hence I do not recommend the paper for publication in its present form. In the following I highlight my main concerns. If these concerns are adequately addressed the paper may become suitable for publication.

The paper discusses errors in the retrieval of aerosol particle "size" from optical measurements. This is justified in the abstract by saying that the "size" AND "size distribution" are fundamental properties of the aerosol, implying that they are two distinct quantities. Obviously, they are not. The word "size" is an ambiguous term for aerosols

whose radii can vary by orders of magnitude. It is not until later in the paper one finds that by size they mean "median radius". But why chose this quantity instead of the effective radius, a commonly used parameter in the aerosol community, defined as the area-weighted radius.

This is not just a matter of personal preference. Many investigators have found that effective radius is a robust measure of aerosol size, since it is less sensitive to the assumed particle size distribution than other parameters, such median or modal radius. Indeed there is no scientific consensus on the median radius of aerosol particles, since microphysical models of aerosols indicate that bulk of the aerosols particles in the stratosphere (possibly more than 90%) are of radii less than 100 nm, to which optical instruments, including in situ optical particle counters, are insensitive. Though these particle are important for the formation of larger particles their effect on solar radiation, and hence on climate is minimal. So, it is not clear in what sense the median radius is a "fundamental" property of aerosols.

The choice of the median radius to define aerosol "size" then leads to the paper's key conclusion that the spectral dependence of aerosol extinction cannot be used to retrieve it with high accuracy. But I am not aware of anyone who has claimed otherwise. While the spectral dependence of aerosol extinction, often condensed into Angstrom Exponent (AE), is a useful size parameter in its one right, using this information one can estimate one of the two parameters of a unimodal lognormal distribution, the modal radius (same as median radius for this distribution) or the width, by prescribing the other parameter a priori. However, it is absurd to claim any scientific validity to either parameter. The primary purpose of doing this is to estimate the effective radius under the assumption that it can be estimated robustly in spite of the inherent ambiguity in the retrieval process. The paper would have been a decent paper if the authors had chosen to focus on this issue.

The authors, however, do discuss errors in the retrieval of other size related parameters that are commonly used by the aerosol community, such as surface area density

(SAD). So this part of the paper is more relevant. But not adequate attention has been paid in discussing the message of the figures 4-6. For example all these figures show a monotonic relationship between particle coarse mode fraction (CMF) and the size related parameters retrieved from extinction AE. Though quantitatively they do not agree with a similar parameter estimated from another distribution, whose parameters are somewhat arbitrarily chosen, it is hard to put lot of significance to this disagreement. One doesn't know, for example what the results would have looked like had they kept the modal radius fixed and had retrieved the width, as is commonly done by the SAGE group. Also the assumption that the modal radius and width of coarse mode particle distribution doesn't vary as the CMF changes is very likely inaccurate. Finally, it would have been very useful if the authors had plotted their calculated relationship between CMF and AE. Since there is a very long history of AE measurements from SAGE, it would have provided some perspective on how often CMF greater than 1%, where errors in the retrieval of various size parameters increase rapidly, may have happened during this record. My guess is that it is quite rare.

Finally, what is most notable from plots 4-6 is the lack of monotonic relationship between CMF and size parameters retrieved from Lidar color ratio. This indicates that LIDAR color ratio doesn't contain useful information about aerosols size, irrespective of how it is defined. This should have been quite apparent had they plotted the relationship between CMF and Lidar color ratio, so there would be no need to do actual retrieval to make the point. Though this wouldn't be a surprise to the various LIDAR groups, this conclusion is important enough to other readers to be highlighted in the abstract.

---

## Referee Comment (RC2) · Anonymous Referee #2 · 30 Oct 2019

**1   General Points**

This paper investigates the bias in retrieved particle size properties between occultation and lidar measurements due to differing sensitivity. In particular, errors due to the assumption of a unimodal lognormal distribution are investigated when the true distribution is bimodal. Error in retrieved distribution parameters as well as integral properties are explored. This paper represents a useful addition to the understanding of bias in retrieved stratospheric aerosol properties, and is clearly and concisely written. I think the discussions and conclusions are at times too broad given the analysis (discussed below) but would recommend publication after minor edits.

[Figure]

In general, the choice of assumed bimodal distribution parameters seems reasonable, however I think some further discussion or analysis of the sensitivity to the choice of these parameters and retrieval algorithm is needed with regards to Figs. 3-6. Without that it is difficult to say exactly how broadly this analysis can be applied. For example, in SAD the increase in error as CMF grows seems reasonable, but then it decreases as CMF grows further. Presumably this is due to the fixed width, and the retrieved median radius of the distribution shifting smoothly between the two modes. But this seems dependent on the retrieval assuming a fixed width and the choice of a priori values and bimodal state. Similarly, do other, commonly retrieved quantities such as effective radius show this effect? How large is the error in the retrieved Lidar extinction?

**2   Specific Points**

*L203: It is to be expected that the systematic differences in retrieved aerosol sizes for lidar and occultation retrievals will increase during periods of enhanced volcanic activity, because then the second particle mode at radii of several hundred nm will be enhanced (e.g., Deshler, 2008).*

- By "size" do the authors mean median radius?  It is not discussed how other size metrics such as effective radius may react, and SAD and volume are not necessarily worse.

- Is this error not dependent on the assumed properties of the retrieval? If the a priori width was too small, could the error not decrease after volcanic activity?

- It is not clear that volcanic activity necessarily leads to an increase in particle size. While true for large eruptions such as Pinatubo, smaller, more recent eruptions have a more ambiguous signal, at least in terms of the wavelength dependence.

*L215-219: It is also important to note that the correct particle size parameters can in principle be retrieved from measurements in any observation geometry (neglecting here issues related to potential non-uniqueness of the solutions), if the assumption of a mono-modal log-normal particle size distribution is correct. . .*

Is this meant to apply to the 2-wavelength retrievals investigated here? If so, I think it should be noted that not only must the unimodal assumption be satisfied, but one size parameter must also be known a priori. If more general, I think non-uniqueness is not an issue that can be neglected.

*L220: The results presented here are also of importance for model simulations of stratospheric aerosols, some of which model aerosol growth processes more or less explicitly*

Is this a recommendation to not compare with derived quantities, but more direct measurements of extinction/back-scatter?

*L83/L201: The effects studied here should, however, also be investigated for aerosol size retrievals from limb-scatter measurements in future studies.*

Some aspects of these effects, and the sensitivity of limb scattering measurements to particle size have been previously investigated by Malinina et al., (2018) and Rieger et al., (2015).

There's a few orphan sentences that should probably be incorporated into surrounding paragraphs or reorganized/expanded (L95, L203, L114, L220).

---

## Author Comment (AC1) · 3 Feb 2020

**Reply to the comment by Alexei Rozanov**

**Comment:** Dear Christian and Christoph,
In your paper you show very interesting results illustrating that a weaker sensitivity of the solar occultation measurements to smaller particles is expected to lead to the overestimation of the mode radius and underestimation of the particle number density if a coarse mode is present but not considered in the retrieval. However, you tend to overgeneralize your conclusions. As shown by Malinina et al. (2019), not mentioned in your paper by the way, the sensitivity of the limb measurements to

smaller particles is very different to that of solar occultation. For this reason I disagree with a blind extension of your findings to the limb geometry as you do, e.g. by writing "Stratospheric aerosol particle size retrievals from satellite limb-scatter measurements can be expected to be affected by similar issues as the occultation and lidar retrievals described here. " To my opinion this statement is unjustified and should be removed.

**Reply: Dear Alexei,**
**we weakened the statement by replacing "can be expected to be" by "may be".**
**However, we don't fully agree with this comment. We agree that there are differences in sensitivities between occultation and limb-scatter measurements, but we see no reason, why limb-scatter measurements should not in principle also be affected by the problem discussed in the paper.**

**We now also cite and briefly discuss the paper by Malinina et al. (2019).**

**Comment:** Furthermore, with the last sentence of the abstract "The results question the overall significance of stratospheric aerosol size retrievals based on optical satellite or lidar measurements, as long as the actual aerosol particle size distribution is not well known." you provide a misleading message to the scientific community. First, based on the results of your paper you can only talk about solar occultation measurements and must not generalize your conclusions to all optical satellite methods, second a known bias in the retrieval products is not yet a reason to question the significance of the retrieval/measurements in general.

**Reply: We agree in part that the statement can be misinterpreted. We agree that we only investigated the effects for lidar and occultation measurements and not for limb-scatter measurements. Still, there is no reason why retrievals based on limb-scatter measurements should not be affected by this problem. We changed**

this sentence and now only make a statement about lidar and occultation mea-
surements. The sentence now reads:

**"The results indicate that stratospheric aerosol size retrievals based on occul-
tation or lidar measurements have to be interpreted with caution, as long as the
actual aerosol particle size distribution is not well known."**

**Comment:** One more technical issue is in the first paragraph of the "Methotology"
section "Aerosol particle size information can in principle be obtained based on
measurements of (a) the spectral dependence of the aerosol extinction or scattering
coefficients (e.g., Yue and Deepak, 1983; Bingen et al., 2003), (b) the scattering phase
function (e.g., Gumbel et al., 2001; Renard et al., 2008), or (c) the polarization of the
radiation scattered by aerosols (e.g., McLinden et al., 1999)." Here you seem to forget
that in limb retrievals the spectral dependence of the radiance rather than that of the
extinction or scattering coefficient is used (Malinina et al., 2018).

**Reply: We agree with this comment. We had implicitly included limb-scatter
measurements in point (a), which is certainly not entirely correct. We now added
limb-scatter measurements explicitly as new point (b).**

---

## Author Comment (AC2) · 3 Feb 2020

**Reply to comments by reviewer 2**

**Comment:** This paper investigates the bias in retrieved particle size properties between occultation and lidar measurements due to differing sensitivity. In particular, errors due to the assumption of a unimodal lognormal distribution are investigated when the true distribution is bimodal. Error in retrieved distribution parameters as well as integral properties are explored. This paper represents a useful addition to the understanding of bias in retrieved stratospheric aerosol properties, and is clearly and concisely written. I think the discussions and conclusions are at times too broad given

the analysis (discussed below) but would recommend publication after minor edits.

**Reply: We thank the reviewer for his/her encouraging comments. As described in detail below, we essentially followed all the suggestions made by the reviewer.**

**Comment:** In general, the choice of assumed bimodal distribution parameters seems reasonable, however I think some further discussion or analysis of the sensitivity to the choice of these parameters and retrieval algorithm is needed with regards to Figs. 3-6. Without that it is difficult to say exactly how broadly this analysis can be applied. For example, in SAD the increase in error as CMF grows seems reasonable, but then it decreases as CMF grows further. Presumably this is due to the fixed width, and the retrieved median radius of the distribution shifting smoothly between the two modes. But this seems dependent on the retrieval assuming a fixed width and the choice of a priori values and bimodal state. Similarly, do other, commonly retrieved quantities such as effective radius show this effect? How large is the error in the retrieved Lidar extinction?

**Reply: We see the reviewer's point and will add further retrieval examples for different sets of PSD parameters to test, how sensitive the results are with respect to the specific input parameters chosen. The results certainly depend to some extent on the assumed input parameters, but the overall conclusions are not affected.**

**We also follow the reviewer's suggestion and test the effect on the effective radius and the retrieved lidar extinction.**

**We would like to point out that we do not claim that the actual PSD is bi-modal. The study investigates, how size retrievals based on an assumed single-model PSD behave if the actual PSD is bi-modal.**

**Comment:** L203: It is to be expected that the systematic differences in retrieved aerosol sizes for lidar and occultation retrievals will increase during periods of enhanced volcanic activity, because then the second particle mode at radii of several hundred nm will be enhanced (e.g., Deshler, 2008).

By "size" do the authors mean median radius? It is not discussed how other size metrics such as effective radius may react, and SAD and volume are not necessarily worse.

**Reply: Yes, we implicitly assumed the median radius to be representative for the overall "size" of the aerosols, which is not entirely correct and may cause confusion. This aspect was also criticized by reviewer 1. We try to avoid this issue by explicitly stating, which representation of aerosol size is meant. These changes – also motivated by the comments of reviewer 1 – affect several parts of the manuscript (changes to the manuscript are highlighed).**

**The reviewer raises a good point, and we now also include results and a discussion on how retrievals of the effective radius are affected.**

**Comment:** Is this error not dependent on the assumed properties of the retrieval? If the a priori width was too small, could the error not decrease after volcanic activity?

**Reply: Yes, we agree that the error will certainly depend to on the assumed properties of the retrieval to some extent. We add some more examples for bimodal distributions with other parameters (median radius and width) to test the overall dependence of the results on the specific input parameters chosen (as also suggested by the reviewer in the previous point).**

**Comment:** It is not clear that volcanic activity necessarily leads to an increase in

particle size. While true for large eruptions such as Pinatubo, smaller, more recent eruptions have a more ambiguous signal, at least in terms of the wavelength dependence.

**Reply: We fully agree with the reviewer. We also have evidence for a temporal decrease in effective radius of stratospheric aerosols after small/moderate eruptions. We now limited the scope of the statement to larger volcanic eruptions, such as, e.g. Pinatubo.**

**Comment:** L215-219: It is also important to note that the correct particle size parameters can in principle be retrieved from measurements in any observation geometry (neglecting here issues related to potential non-uniqueness of the solutions), if the assumption of a mono-modal log-normal particle size distribution is correct ... Is this meant to apply to the 2-wavelength retrievals investigated here? If so, I think it should be noted that not only must the unimodal assumption be satisfied, but one size parameter must also be known a priori. If more general, I think non-uniqueness is not an issue that can be neglected.

**Reply: We agree with the reviewer that not only the unimodal assumption must be correct, but also one of the size parameters must be known. We also agree that the non-uniqueness of the solution is a general issue, which is particularly important in the expected radius range for the lidar measurements (see the discussion in Zalach et al., AMTD, 2018). We adjusted the statements to become more precise and added some additional text.**

**Comment:** L220: The results presented here are also of importance for model simulations of stratospheric aerosols, some of which model aerosol growth processes more

or less explicitly

Is this a recommendation to not compare with derived quantities, but more direct measurements of extinction/back-scatter?

**Reply: This is a very good point, which we had not thought about. Our intention was to point out potential problems if models are verified or validated using stratospheric aerosol size information, e.g. retrieved from solar occultation measurements, which is sometimes done (e.g. in Hommel, 2008). We expanded this sentence. We think the comparison of all quantities (size information and extinction etc.) is important and should be carried out.**

**Comment:** L83/L201: The effects studied here should, however, also be investigated for aerosol size retrievals from limb-scatter measurements in future studies.

Some aspects of these effects, and the sensitivity of limb scattering measurements to particle size have been previously investigated by Malinina et al., (2018) and Rieger et al., (2015).

**Reply: The reviewer is again correct and we included references to the two papers, as well as short descriptions.**

**Comment:** There's a few orphan sentences that should probably be incorporated into surrounding paragraphs or reorganized/expanded (L95, L203, L114, L220).

**Reply: Thanks for pointing this out. The sentence in line 95 is now combined with the following paragraph (that was actually our intention). The sentence in line 114 was combined with the previous paragraph. The sentence in line 203 was extended to a short paragraph based on the above comments by the**

**reviewer. Similarly, line 220 was extended following the reviewer's suggestion.**

**References:**

Hommel, R.: Die Variabilität von stratosphärischem Hintergrund-Aerosol. Eine Untersuchung mit dem globalen sektionalen Aerosolmodell MAECHAM5-SAM2., Ph.D. thesis, Universität Hamburg, 2008.
* * *

---

## Author Comment (AC3) · 3 Feb 2020

**Reply to comments by reviewer 1**

**Comment:** The paper contains some useful information. But overall content of the paper is weak and would be of limited interest to the researchers in the field. Hence I do not recommend the paper for publication in its present form. In the following I highlight my main concerns. If these concerns are adequately addressed the paper may become suitable for publication.

[Figure]

**Reply: We thank the reviewer for his/her comments, and we tried to address all the points raised in an appropriate way.**

**Comment:** The paper discusses errors in the retrieval of aerosol particle "size" from optical measurements. This is justified in the abstract by saying that the "size" AND "size distribution" are fundamental properties of the aerosol, implying that they are two distinct quantities. Obviously, they are not. The word "size" is an ambiguous term for aerosols whose radii can vary by orders of magnitude. It is not until later in the paper one finds that by size they mean "median radius". But why chose this quantity instead of the effective radius, a commonly used parameter in the aerosol community, defined as the area-weighted radius.

**Reply: We appreciate the reviewer's comment and fully agree that "size" is an ambiguous term. We have adjusted the manuscript such that is clear from the beginning what "size" refers to. We now also include results showing how retrievals of the effective radius behave for occultation and lidar geometries if the coarse mode fraction is varied, following the reviewer's suggestion.**

**We would like to point out that we do not claim (and did not do that in the paper) that the median radius is an appropriate measure of aerosol size. The reason we use it is that it has been used in previous studies, e.g., on stratospheric aerosol particle size retrievals from the SAGE II solar occultation instruments. The main motivation of the paper was to investigate a potential reason for the finding, that aerosol particle size retrievals from solar occultation measurements seem to yield systematically larger values than retrievals from other measurements. This is essentially independent of the specific measure of aerosol size used (e.g., median, mode or effective radius). We included statements in the manuscript clarifying that we do not claim that some of the assumptions made are fully correct.**

**Comment:** This is not just a matter of personal preference. Many investigators have found that effective radius is a robust measure of aerosol size, since it is less sensitive to the assumed particle size distribution than other parameters, such median or modal radius. Indeed there is no scientific consensus on the median radius of aerosol particles, since microphysical models of aerosols indicate that bulk of the aerosols particles in the stratosphere (possibly more than 90instruments, including in situ optical particle counters, are insensitive. Though these particle are important for the formation of larger particles their effect on solar radiation, and hence on climate is minimal. So, it is not clear in what sense the median radius is a "fundamental" property of aerosols.

**Reply: We are well aware of the issue that optical measurements are insensitive of the smallest particles and this is exactly an important aspect of the study. The reviewer's comment touches on some fundamental issues, which cannot be resolved with optical measurements. In order to retrieve particle size information (whatever assumptions are made) we need to assume a particle size distribution. This is also the case, if we retrieve the effective radius. If we assume a mono-modal log-normal distribution, then effective radius is related to the median radius and the distribution width via an analytical relationship. In other words, there is not a really fundamental difference between retrieving the effective radius and the median radius (assuming a width parameter).**

**However, we do follow the reviewer's suggestion and now also test the effects of a changing coarse mode fraction on the effective radius and think this is a very good idea.**

**We also want to point out that we do not claim that the median radius is a key fundamental property of the aerosol. The particle size distribution is a funda-**

**mental property in our opinion. However, we do not know what the actual size distribution is, and we will probably never know it exactly for any given case. We are well aware that the log-normal particle size distribution is a model, which will very likely differ from the actual size distribution in any given case. However, this is not a key point of the study. The main point of the paper is to investigate, how aerosol size retrievals based on a mono-modal size distribution behave, if the actual size distribution is bi-modal. We even state explicitly on page 7 of the paper: "We would like to point out that we do not claim that the actual particle size distribution of stratospheric aerosols is a bi-modal log-normal distribution.".**

**Comment:** The choice of the median radius to define aerosol "size" then leads to the paper's key conclusion that the spectral dependence of aerosol extinction cannot be used to retrieve it with high accuracy. But I am not aware of anyone who has claimed otherwise. While the spectral dependence of aerosol extinction, often condensed into Angstrom Exponent (AE), is a useful size parameter in its one right, using this information one can estimate one of the two parameters of a unimodal lognormal distribution, the modal radius (same as median radius for this distribution) or the width, by prescribing the other parameter a priori. However, it is absurd to claim any scientific validity to either parameter. The primary purpose of doing this is to estimate the effective radius under the assumption that it can be estimated robustly in spite of the inherent ambiguity in the retrieval process. The paper would have been a decent paper if the authors had chosen to focus on this issue.

**Reply: We agree with the reviewer that the use of "aerosol size" is not very precise and that the median radius is not a good descriptor of aerosol size. But the appropriateness of the median radius as a descriptor of the aerosol size is not the main point of the paper. We carried out these synthetic retrievals of the median radius, because (a) this has been done in previous studies, e.g., based**

on solar occultation measurements, and (b) we wanted to investigate a potential reason for the essentially systematic high bias in median radii retrieved from these occultation measurements.

We now also test how retrievals of the effective radius from two-colour lidar and occultation measurements depend on the coarse mode fraction. Furthermore, we show the Angstrom exponents as a function of coarse mode fraction.

**Comment:** The authors, however, do discuss errors in the retrieval of other size related parameters that are commonly used by the aerosol community, such as surface area density(SAD). So this part of the paper is more relevant. But not adequate attention has been paid in discussing the message of the figures 4-6. For example all these figures show a monotonic relationship between particle coarse mode fraction (CMF) and the size related parameters retrieved from extinction AE. Though quantitatively they do not agree with a similar parameter estimated from another distribution, whose parameters are somewhat arbitrarily chosen, it is hard to put lot of significance to this disagreement. One doesn't know, for example what the results would have looked like had they kept the modal radius fixed and had retrieved the width, as is commonly done by the SAGE group. Also the assumption that the modal radius and width of coarse mode particle distribution doesn't vary as the CMF changes is very likely inaccurate. Finally, it would have been very useful if the authors had plotted their calculated relationship between CMF and AE. Since there is a very long history of AE measurements from SAGE, it would have provided some perspective on how often CMF greater than 1

**Reply: We thank the reviewer for these useful comments. We realize that the chosen parameters may not be representative. We now test a few more cases to investigate, how dependent the retrieval results are on the a priori assumptions.**

**In addition, we also show the dependence of the AE on CMF, as suggested by the reviewer.**

**Comment:** Finally, what is most notable from plots 4-6 is the lack of monotonic relationship between CMF and size parameters retrieved from Lidar color ratio. This indicates that LIDAR color ratio doesn't contain useful information about aerosols size, irrespective of how it is defined. This should have been quite apparent had they plotted the relationship between CMF and Lidar color ratio, so there would be no need to do actual retrieval to make the point. Though this wouldn't be a surprise to the various LIDAR groups, this conclusion is important enough to other readers to be highlighted in the abstract.

**Reply: Looking at Fig. 3, there is not a non-monotonic dependence between CMF and the median radius, but a monotonic one. We would like to point out that the non-monotonic relationship between volume density/surface area density and CMF results from the non-monotonic relationship between number density and CMF (Fig. 4), not from the relationship between median radius and CMF. The median radius increases monotonically with CMF in the CMF-range considered. The same is true for the effective radius (not shown in the paper). It is therefore not correct to state that the lidar color ratio contains no useful information on aerosol size.**

**We included a more detailed discussion of these aspects in the paper and hope this appropriately addresses the reviewer's concerns.**